# Mechanical Properties and Energy Absorption Abilities of Diamond TPMS Cylindrical Structures Fabricated by Selective Laser Melting with 316L Stainless Steel

**DOI:** 10.3390/ma16083196

**Published:** 2023-04-18

**Authors:** Dorota Laskowska, Tomasz Szatkiewicz, Błażej Bałasz, Katarzyna Mitura

**Affiliations:** Faculty of Mechanical Engineering, Koszalin University of Technology, Śniadeckich 2, 75-620 Koszalin, Poland; blazej.balasz@tu.koszalin.pl (B.B.); katarzyna.mitura@tu.koszalin.pl (K.M.)

**Keywords:** additive manufacturing, selective laser melting, lattice structures, triply periodic minimal surface, lightweight metallic structures, quasi-static compression, energy absorption

## Abstract

Triply periodic minimal surfaces (TPMS) are structures inspired by nature with unique properties. Numerous studies confirm the possibility of using TPMS structures for heat dissipation, mass transport, and biomedical and energy absorption applications. In this study, the compressive behavior, overall deformation mode, mechanical properties, and energy absorption ability of Diamond TPMS cylindrical structures produced by selective laser melting of 316L stainless steel powder were investigated. Based on the experimental studies, it was found that tested structures exhibited different cell strut deformation mechanisms (bending-dominated and stretch-dominated) and overall deformation modes (uniform and “layer-by-layer”) depending on structural parameters. Consequently, the structural parameters had an impact on the mechanical properties and the energy absorption ability. The evaluation of basic absorption parameters shows the advantage of bending-dominated Diamond TPMS cylindrical structures in comparison with stretch-dominated Diamond TPMS cylindrical structures. However, their elastic modulus and yield strength were lower. Comparative analysis with the author’s previous work showed a slight advantage for bending-dominated Diamond TPMS cylindrical structures in comparison with Gyroid TPMS cylindrical structures. The results of this research can be used to design and manufacture more efficient, lightweight components for energy absorption applications in the fields of healthcare, transportation, and aerospace.

## 1. Introduction

Triply periodic minimal surfaces (TPMS) are structures inspired by nature with periodically, infinite continuous non-self-intersecting surfaces with zero mean curvature in three independent directions [1,2,3]. Thanks to the mathematical description, it is possible to arbitrarily change the global or regional porosity (relative density) [4,5,6,7], unit cell shape, size, and arrangement [8,9,10,11], which allows for a global or regional change in the mechanical properties and creates new structural designs for innovative applications. The most important features of TPMS are the elimination of the effect of stress concentration [12]. Thanks to this, TPMS performs better mechanical properties and absorption energy capacity in comparison to strun structures, such as octet, cubic, body center cubic (BCC), or face-centered cubic (FCC) [12,13,14]. Due to the large volume-specific surface area, TPMS structures can be used as radiators [15,16,17], chemical microreactors [18,19], and membranes [20,21]. Thanks to the system of open internal channels, which allows multiple reflections of waves, the TPMS structures have been used in the absorption of acoustic waves [22,23] and electromagnetic microwave absorption [24]. Numerous studies confirm the feasibility of using TPMS structures in the design of new bone implant systems [25,26,27,28,29,30]. In addition, TPMS structures can be the basis for designing new solutions in mechanical energy absorption [31,32,33,34,35].

Most of the research has focused on the classic cubic arrangement of skeletal or sheet TPMS unit cells with uniform or graded relative density. In the case of a skeletal unit cell, the design space is divided into two subspaces (void and material) separated by a smooth and nonintersecting surface. In the case of the sheet unit cell, two surfaces are generated in the design space that is shifted relative to each other, which divides the space into two unconnected void areas and a material area. Research carried out, among others, by Li et al. [4], Zhou et al. [5], and Zhang et al. [36] show that sheet TPMS performed better mechanical properties, strength, and energy absorption capacity compared to skeletal TPMS. Maskery et al. [37] investigated the energy absorption capacity of gyroid, diamond, and primitive TPMS sheet structures manufactured by selective laser sintering from PA2200. Experimental results confirm that the geometry of the elementary cell has the greatest influence on the mechanical response and deformation mechanism. The porosity and surface area for every unit cell decreases with increasing wall thickness. These observations were also confirmed by Zhang et al. [33], Bobber et al. [38], and Novak et al. [39] for structures manufactured by selective laser melting from 316L stainless steel and Ti6Al4V. Moreover, most authors conclude on the advantage of diamond structures over other TPMS in terms of mechanical properties and energy absorption capacity. Kladovasilakis et al. [40] reported that among the tested TPMS structures (manufactured by selective laser melting from polyamide 12), the highest energy absorption was observed for the diamond structure. On the other hand, the lowest energy absorption per volume unit was observed for primitive structures. The research carried out by Sokollu et al. [41] showed that the diamond structure had the highest ultimate tense strength and yield stress, while the gyroid structure had the highest energy absorption capabilities. The properties of diamond structures have been student extensively by Wang et al. [42], Al-Ketan et al. [43], and Chen et al. [44].

So-called cylindrical lattice structures (CLS) are most commonly used in energy absorption applications. The CLS consists of ribs in a circumferential and spiral direction, and the intersection of ribs forms periodic patterns [45]. Reducing the weight in combination with good mechanical properties (height stiffness-to-weight characteristics) made CLS widely used in transport, aerospace, and cosmonautics [45,46,47,48]. The rapid development of additive manufacturing technology has recently increased interest in the use of TPMS in CLS design. Wang et al. [49] proposed a mapping methodology that allows the designing of TPMS structures with cylindrical arrangements of unit cells by describing them with polar coordinates. Based on the fine element analysis, it was observed that the gyroid cylindrical lattice structure (G-CLS) shows improved energy absorption capability in comparison to CLS based on the strun unit cells. Research carried out by Cao et al. for G-CLS [50] and primitive cylindrical lattice structure (P-CLS) [51] showed that geometrical defects have a significant impact on the mechanical properties, what should be taken into consideration during analyzing the deformation mode and preparing simulation models. Szatkiewcz et al. [52] analyzed the mechanical properties, energy absorption capacity, and deformation models of Gyroid TPMS cylindrical structures. The experimental results show the relationship between the selected structural parameters, mechanical properties, and energy absorption abilities. It was observed that the tested structures represented two different modes of deformation, with the layer-by-layer mode being dominant.

This study investigated the Diamond TPMS cylindrical structures fabricated by selective laser melting with 316L stainless steel powder. Based on the experimental study, compressive behavior, overall deformation mode, mechanical properties, and energy absorption ability are evaluated. Next, the results were compared and discussed with our previous work [52] concerning Gyroid TPMS cylindrical structures. The objective was an evaluation of the possibility of tested structures in energy absorption applications.

## 2. Materials and Methods

### 2.1. Diamond Lattice Structure Design

Diamond (Figure 1) is one type of TPMS, which can be designed by defining the isosurface (D = 0) equation:D = [sin(k_x_x)sin(k_y_y)sin(k_z_z) + sin(k_x_x)cos(k_y_y)cos(k_z_z) + cos(k_x_x)sin(k_y_y)cos(k_z_z) + cos(k_x_x)cos(k_y_y)sin(k_z_z)]^e^ − t^e^,(1)
k_i_ = 2π * u_i_/L_i_  i = x,y,z,(2)
where:k_i_—TPMS function periodicity;u_i_—number of unit cells in x, y, and z dimension;L_i_—absolute size of the porous structure in x, y, and z direction;e—exponent, which determines the type of unit cell: skeletal (for e = 1) and sheet (for e = 2);t—parameter, which determines the parts of the volume of the regions separated by isosurface [4,37,52,53].

In this study, the lattice structures with the cylindrical arrangement of unit cells were designed with nTopology 3.23.3 (nTopology, New York, NY, USA) software [54] based on the methodology described by Wang et al. [49] and Szatkiewicz et al. [52]. The geometrical dimension (Figure 2) and relative density (20%) of the structures adopted are the same as in our previous studies to enable comparative analysis. The plan of the experiment assumed the design and manufacture of a series of 9 structures with 12, 9, and 6 cells in a circumferential direction (n_circum_), 1, 1.5, and 2 cells in the radial direction (n_radial_), and 3 cells layer in an axial direction (n_axial_). The symbol of each structure consisted of 4 parts, respectively: the type of TPMS topology, n_circum_, n_radial_, and the wall thickness.

### 2.2. Metal Powder Characterization and Sample Structures Fabrication

The sample structures were produced using MetcoAdd^TM^ 316L-A (Oerlikon Metco Inc., Troy, MI, USA) 316L austenitic stainless steel powder. The chemical composition of the powder is shown in Table 1. The morphology of the powder (Figure 3A) was determined using a PHENOM PRO scanning electron microscope (Thermo Fisher Inc., Waltham, MA, USA).

Laser particle sizer ANALYSETTE 22 MicroTec Plus (Fritsch GmbH, Amberg, Germany) was used to determine the particle size distribution of the powder (Figure 3B) in accordance with PN-ISO 9276-1 [55].

**Table 1 materials-16-03196-t001:** Chemical composition of MetcoAdd^TM^ 316L-A powder [56].

Element	Fe	Cr	Ni	Mo	C	Other
**Weight percent (%)**	Balance	18	12	2	<0.03	<1.0

All samples were fabricated using laser sintering technology with the same process parameters shown in Table 2 using the SLM ORLAS CREATOR^®^ process equipment (OR Lasertechnologie GmbH, Dieburg, Germany).

The greatest advantage of porous structures based on the TPMS topography is their self-supporting nature. This allows the fabrication of a structure with open pores and channels using a limited support system. The supports were removed mechanically. The fabricated structures were then cleaned of unbound powder and then subjected to a wash process in an ultrasound washer in distilled water for 10 min. To determine the difference between the actual (m_r_) and design (m_d_) weight, each sample was weighed on a precision balance with an accuracy of d = 0.01 g. The designed mass was calculated by the equation:m_d_ = ρ_S_ ∗ V_L_,(3)
where V_L_ is the volume of the structure calculated from the CAD design, and ρ_S_ is the density of the sample produced from 316L powder with the same technological parameters as the tested structures. In this study, ρ_S_ was set as 7.62 g/cm^3^. Based on this, it was calculated that the designed mass for tested samples should be 16.57 g. The quasi-static compression test was performed for structures in which the difference between the design and the real mass was about 2%. Figure 4 shows the collection of all Diamond TPMS cylindrical structures that were fabricated to perform the tests. In addition, Figure 5 shows their vertical cross-sections. Table 3 presents the exact specification of the tested samples.

### 2.3. Quasi-Static Compression Test

A quasi-static compression test was carried out to determine the mechanical properties of the porous structures. Tests were carried out on a Zwick Z400E (ZwickRoell GmbH and Co., Ulm, Germany) testing machine in accordance with ISO 133:14 standard [57] with continuous loads at a rate of 2 mm/min (this is about 8.33% strain per minute) at ambient temperature. The loading direction was parallel to the building direction. For each design of the structure design, 3 tests were carried out. The compression tests were recorded using the camera to analyze the tested structure’s deformation models.

Based on the uniaxial stress–strain curve, the elastic modulus (E_L_), yield strength (σ_y_), plateau stress (σ_L_), and energy absorption capacity (W_V_) were determined. The modulus of elasticity was determined from the angle of inclination of the linear elastic section of the stress–strain curve, and the yield strength was determined using the 0.2% offset method. Plateau stress was defined as the arithmetic mean of the stress in the strain intervals from 20% to 30% of the compressive strain according to ISO 13314:2011.

The energy absorption capacity was defined as the area under the stress–strain curve (energy absorption per unit volume):W_V_ = ∫σ(ε)dε.(4)

The energy absorption efficiency [4,6,49,56] was defined as the quotient of the energy absorption and the highest stress obtained for compression up to the strain ε, according to the equation:η(ε) = (1/σ(ε)) ∗ W_V_.(5)

## 3. Results

### 3.1. Compression Process and Overall Deformation Model

The course of change in the stress–strain diagrams from quasi-static compressive tests for tested structures is presented in Figure 6, Figure 7 and Figure 8. All curves exhibit three characteristic sections: a linear elastic section, followed by a long plateau section, and ended by a densification section. For the linear elastic section, the compressive stress increased rapidly until the struts yielded due to bending or stretching. The shape of the plateau section is the result of the sequential collapse of the unit cells because of bucking, brittle crushing, or yielding, depending on the type of construction material and unit cell geometry. After crossing the so-called densification point (ε_D_), the stress value rises due to the entire collapsing of unit cell walls and reaching contacts. The point of densification is, therefore, the limit of a structure’s suitability for energy absorption applications [4,6,52,57,58,59]. In some cases, a linear-elastic section was preceded by a non-linear stage, the reason for which was the lack of full contact between the specimen surface and the testing machine head [28,52].

According to Gibson et al. [57] and Ashby [60] works, depending on the deformation mechanism of the cell strut and the overall deformation models, lattice structures can be divided into stretch-dominated and bending-dominated structures. The stress–strain curve of stretch-dominated structures is characterized by an initial increase in stress followed by a rapid decrease in stress and a series of fluctuations in the course of the plateau section associated with the progressive collapse of the unit cells layers. In comparison, the bending-dominated structures exhibit a more constant curse of plateau section. Based on this, the tested Diamond TPMS cylindrical structures with n_radial_ = 1 were classified as bending-dominated lattice structures, while the structures with n_radial_ = 1.5 and n_radial_ = 2 were classified as stretch-dominated lattice structures.

In order to illustrate the overall deformation mode of Diamond TPMS cylindrical structures during the compression test, camera images were recorded. Based on the analysis of the images, it was found that the studied structures were destroyed according to two different deformation modes. For bending-dominated structures, horizontal bending of the walls of unit cells located in the middle layers of the structure was observed. This gave the structure a characteristic barrel shape (the diameter of the structure after the compression test was higher than before). With the increase in strain, the folding and collapse of the walls of unit cells located in the upper or lower layers of the structure were observed. This so-called uniform deformation mode was presented for structures with n_radial_ = 1 (Figure 9). The second mode of deformation is characterized by the successive collapse of the walls of elementary cells until the structure is completely destroyed. This is the so-called “layer-by-layer” deformation mode, which was presented for structures with n_radial_ = 1.5 and n_radial_ = 2 (Figure 10).

### 3.2. Compressive Mechanical Properties

Compressive mechanical properties (elastic modulus, yield strength, and plateau stress) of the tested structures are summarized in Table 4. Generally, it was observed that increasing the number of unit cells in radial (n_radial_) and circumferential (n_circum_) directions reduced the value of elastic modulus. For the stretch-dominated structures increasing n_circum_ increased the yield strength and plateau stress. The bending-dominated structures showed similar values of yield strength and significantly higher values of plateau stress. Generally, increasing n_radium_ decreased the value of plateau stress for all tested structures. The numerical results are also presented in Figure 11 to better illustrate the described relationships.

### 3.3. Energy Absorption

The energy absorption ability is a significant evaluation index of the lattice structures for energy absorption applications. Figure 12A,C,E shows the cumulative energy absorption per unit volume as a function of compressive strain. It can be found that energy absorption gradually and linearly increased until the densification point was reached. After that, an exponential increase in energy absorption was observed due to the entire collapsing of unit cell walls and reaching contacts. For stretch-dominated structures (n_radial_ = 1 and n_radial_ = 1.5), the densification point was reached for lower values of compressive stresses.

Figure 12B,D,F presents the efficiency–strain curve. It can be observed that energy absorption efficiency gradually increased until the densification point was reached. For stretch-dominated structures (n_radial_ = 1 and n_radial_ = 1.5) in the raising stage of the curve, the fluctuation was observed. It can be explained that the energy absorption efficiency is related to fluctuations in stress values caused by the collapse of the cell walls of elementary structures. The absence of oscillatory changes in the phase of the absorbed energy gain indicates the stability of the absorption energy process during compression for the studied structures [52,56,60].

The basic energy absorption parameters of tested Diamond TPMS cylindrical structures are summarized in Table 5. Generally, the energy absorption up to the densification point decreased with increasing the number of unit cells in the radial direction (n_radial_). The highest value of the energy absorption up to the densification point (17.53 MJ/m^3^) was observed in the case of Diamond_9_1_0.64, which was a bending-dominated structure. The lowest value (7.67 MJ/m^3^) was shown for Diamond_6_2_0.58, which was a stretch-dominated structure. This confirms the advantage of structures with a long and flat plateau in energy absorption applications [37]. The data are also presented in Figure 13 to better illustrate the described dependencies.

## 4. Discussion

The research results described in this paper were compared with research on mechanical properties and energy absorption abilities of Gyroid TPMS cylindrical structures, which were described in one of our previous works [52]. This was possible due to the same experiment methodology.

Based on the stress–strain curve, it was found that all Gyroid TPMS cylindrical structures were bending-dominated structures, while this compressive behavior was observed only for Diamond TPMS cylindrical structures with 1 unit cell in the radial direction. Therefore, different relationships between design parameters (n_circum_ and n_radial_) and compressive mechanical properties (E_L_, σ_y_, σ_pl_) were observed, which confirmed that mechanical and compressive behavior depends not only on the relative density but also on the geometry and arrangement of unit cells. These relationships are summarized in Table 6.

**Table 6 materials-16-03196-t006:** Relationships between design parameters and mechanical properties for Gyroid [52] and Diamond TPMS cylindrical structures. Where: ↑ increase. ↓ decrease.

	Gyroid TPMS Cylindrical Structures	Diamond TPMS Cylindrical Structures	Figure
Elastic modulus	When n_radial_ ↑, then E_L_ ↓ for all structures When n_circum_ ↑, then E_L_ ↑ for all structures	When n_radial_ ↑, then E_L_ ↑ for all structures When n_circum_ ↑, then E_L_ ↑ for all structures	Figure 14A–C
Yield strength	When n_radial_ ↑, then σ_y_ ↓ f for all structures. When n_circum_ ↑, then σ_y_ ↑ for all structures	When n_radial_ ↑, then σ_y_ ↓ for stretch-dominated structures When n_circum_ ↑, then σ_y_ ↑ for stretch-dominated structures σ_y_ similar for all bending-dominated structures	Figure 15A–C
Plateau stress	When n_radial_ ↑, then σ_pl_ ↓ for all structures When n_circum_ ↑, then σ_pl_ ↑ for all structures	When n_radial_ ↑, then σ_pl_ ↓ for all structures When n_circum_ ↑, then σ_pl_ ↑ for stretch-dominated structures When n_circum_ ↑, then σ_pl_ ↓ for bending-dominated structures	Figure 16A–C

The energy absorption ability is a significant evaluation index of the lattice structures for energy absorption applications. Comparing the basic energy absorption parameters for Gyroid and Diamond TPMS cylindrical structures, the following was found:In both cases, the structure that absorbed the highest value of energy to the densification point had n_circum_ = 9 and n_radial_ = 1;In both cases, the structure that absorbed the lowest value of energy to the densification point had n_circum_ = 6 and n_radial_ = 2.

Therefore, the comparative analysis was extended by another evaluation criterion, which is the maximum efficiency of energy absorption. It is energy absorption efficiency for the stress corresponding to the densification point read from the curve efficiency–strain. The data are summarized in Table 7.

Comparing results for structures with n_radial_ = 1, it was found that Diamond type structures absorb more energy in comparison to Gyroid-type structures (Figure 17), but their maximum efficiency was about 2–3% lower. Comparing the courses of the stress–strain or efficiency–strain curves for this group, it can be concluded that Diamond type structures presented more stability of energy absorption during compression.

The stretch-dominated deformation mechanism of the cell strut characterized diamond TPMS cylindrical structures with n_radial_ = 1.5 and n_radial_ = 2. Therefore, they absorbed much less energy than Gyroid TPMS cylindrical structures with the same number of n_radial_.

## 5. Conclusions

This study investigated the mechanical properties and energy absorption capacity of Diamond TPMS cylindrical structures fabricated in the additive manufacturing process by selective laser melting of 316L stainless steel powder. Based on the stress–strain curve, the tested Diamond TPMS cylindrical structures were categorized as stretch-dominated (n_radial_ = 1.5 and n_radial_ = 2) and bending-dominated (n_radial_ = 1). The dependence of the mechanical properties on the geometric parameters of the designed structures over the entire range of changes in the values of these parameters was observed. The following were found:An increase in n_circum_ and n_radial_ causes an increase in the value of E_L_ for all structures;For stretch-dominated structures, an increase in n_circum_ and a decrease in n_radial_ causes an increase in the value of σ_y_;An increase in n_circum_ causes an increase in the value of σ_pl_ for stretch-dominated structures and a decrease in the value of σ_pl_ for bending-dominated structures;

These relationships can be traced in Figure 11 in Table 6.

An ideal energy absorber should accommodate deformations at almost constant stress and thus have a long and flat plateau section. The amount of energy absorbed up to the densification point is determined by the area under the stress–strain curve, so the highest possible plateau stress value is preferred. Such features are exhibited by structures dominated by bending. On the other hand, fluctuations in the plateau region, typical of stretch-dominated structures, limit the amount of energy absorbed [58,61,62,63]. Evaluation of basic absorption parameters shows the advantage of bending-dominated Diamond TPMS cylindrical structures in absorption energy applications in comparison with stretch-dominated Diamond TPMS cylindrical structures. However, their elastic modulus and yield strength were lower.

Comparative analysis showed a slight advantage of cylindrical Diamond TPMS structures with bending-dominated cellular strut deformation mechanisms for energy absorption applications. This type of structure absorbed more energy compared to Gyroid-type structures and was characterized by greater stability of energy absorption during compression. However, all cylindrical Gyroid TPMS structures that have been studied previously have been dominated by this deformation mechanism, allowing for more design freedom.

## Figures and Tables

**Figure 1 materials-16-03196-f001:**
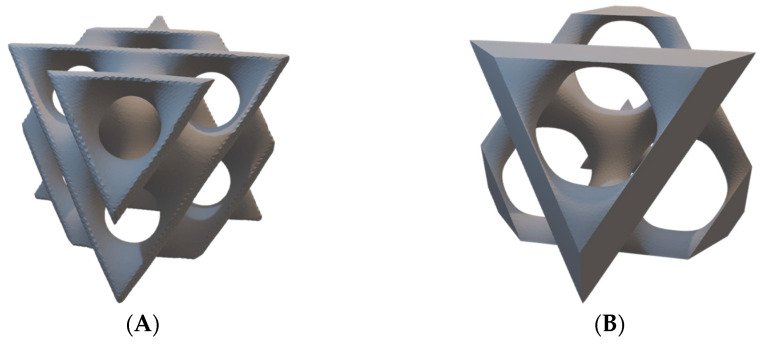
Three-dimensional models of Diamond unit cell: (**A**) sheet (matrix, shell); (**B**) skeletal (network).

**Figure 2 materials-16-03196-f002:**
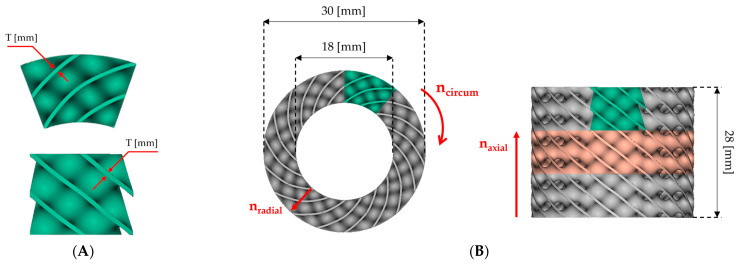
Project of Diamond TPMS cylindrical structures on example Diamond_9_1_0.64 structure: (**A**) unit cell (top and side view); (**B**) geometrical dimensions (top and side view; green color- single unit cell; orange color- single layer of unit cells).

**Figure 3 materials-16-03196-f003:**
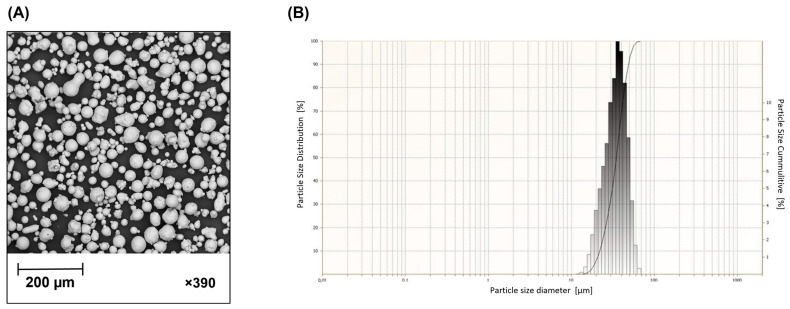
MetcoAdd^TM^ 316L-A powder characterization: (**A**) SEM morphology, (**B**) particle size distribution analysis.

**Figure 4 materials-16-03196-f004:**
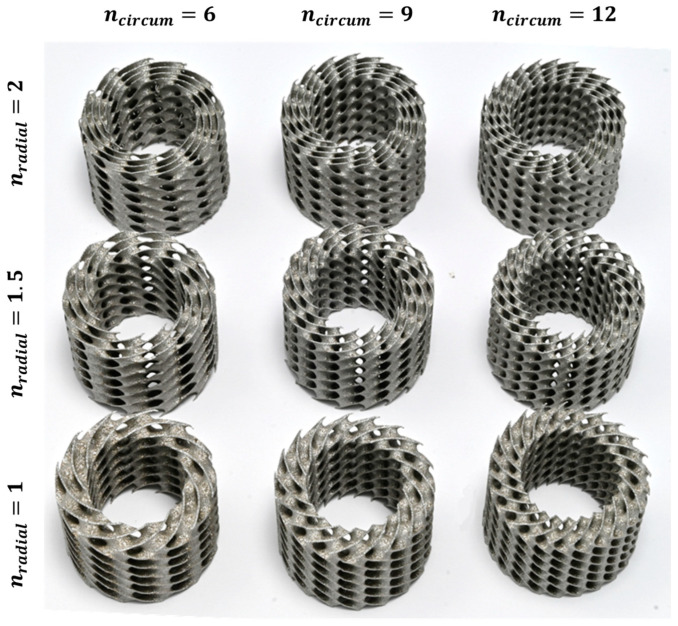
Diamond TPMS cylindrical structures fabricated by selective laser melting (SLM) of 316L stainless steel powder used in the study.

**Figure 5 materials-16-03196-f005:**
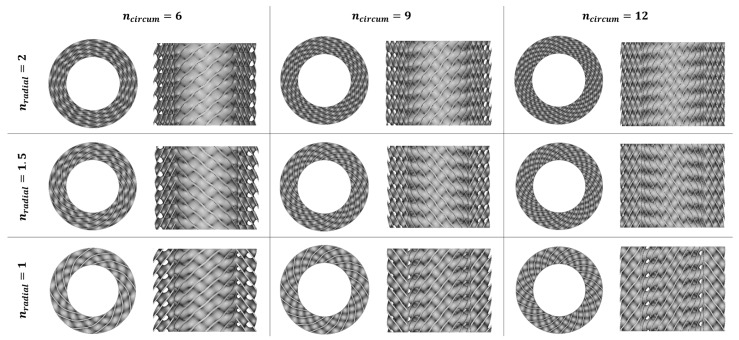
CAD models of Diamond TPMS cylindrical structures—top view and vertical cross-section.

**Figure 6 materials-16-03196-f006:**
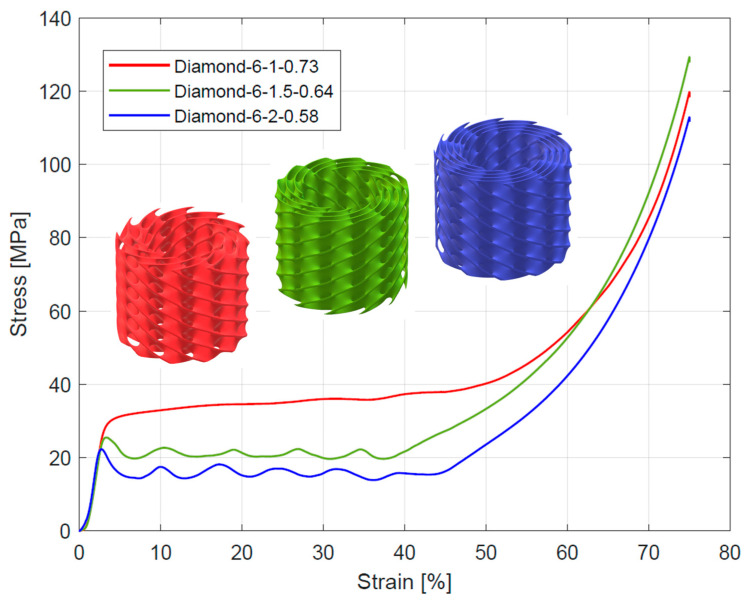
Stress–strain curves from compressive testing of Diamond TPMS cylindrical structures with 6 unit cells in a circumferential direction.

**Figure 7 materials-16-03196-f007:**
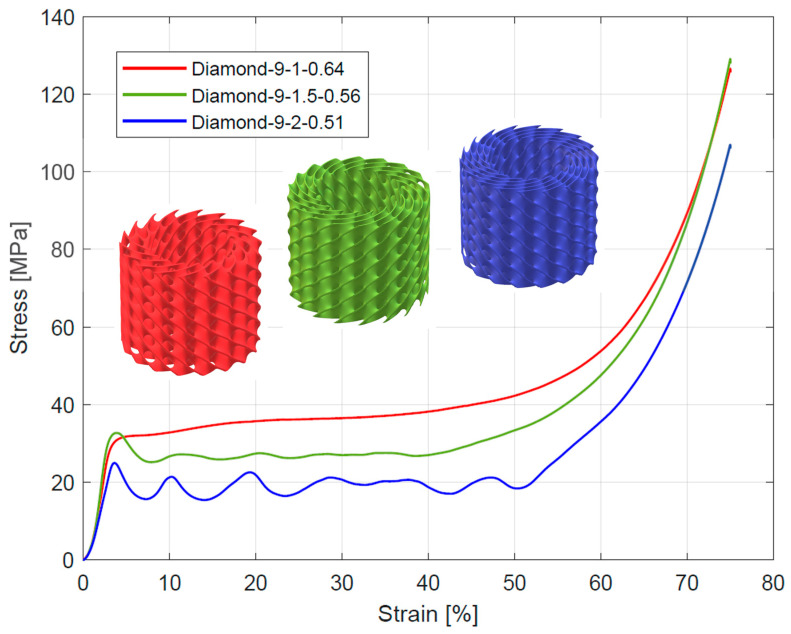
Stress–strain curves from compressive testing of Diamond TPMS cylindrical structures with 9 unit cells in a circumferential direction.

**Figure 8 materials-16-03196-f008:**
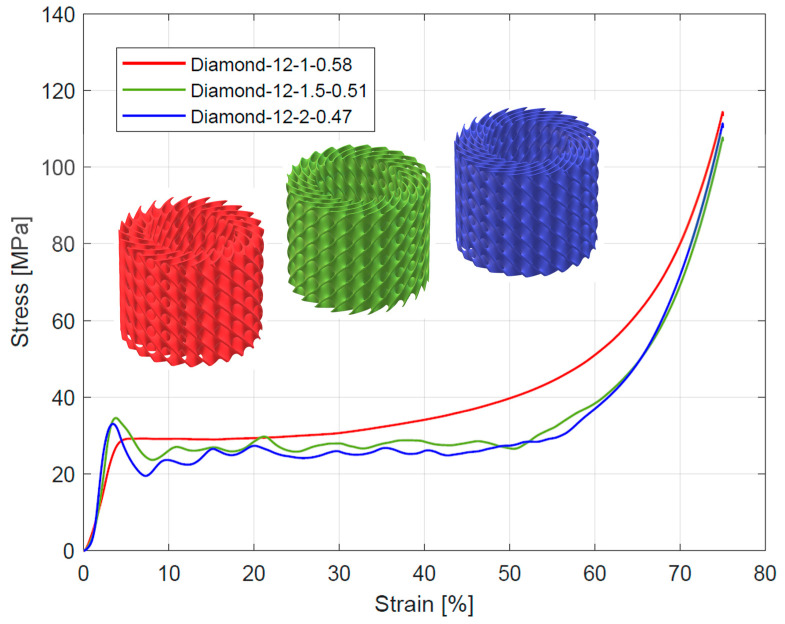
Stress–strain curves from compressive testing of Diamond TPMS cylindrical structures with 12 unit cells in a circumferential direction.

**Figure 9 materials-16-03196-f009:**
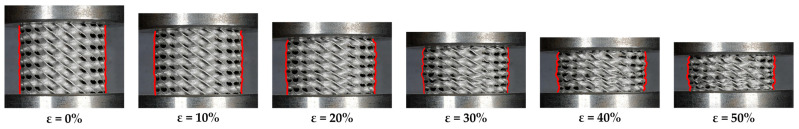
Uniform deformation mode at levels of strain 0%, 20%, 30%, 40%, and 50% for bending-dominated structure on the example of Diamond_9_1_0.64.

**Figure 10 materials-16-03196-f010:**
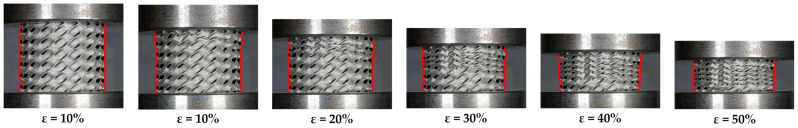
“Layer-by-layer” deformation mode at levels of strain 0%, 20%, 30%, 40%, and 50% for bending-dominated structure on the example of Diamond_9_1.5_0.51.

**Figure 11 materials-16-03196-f011:**
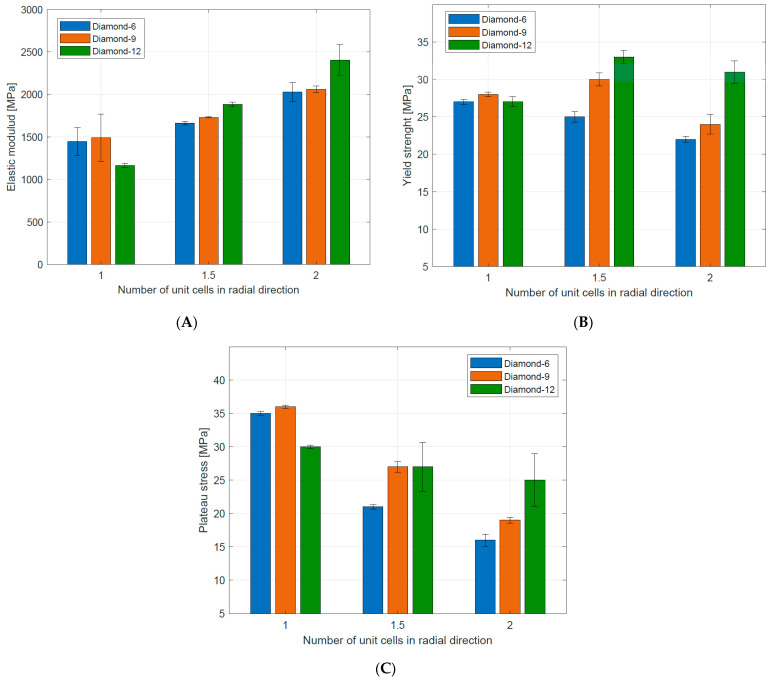
Mechanical properties of tested Diamond TPMS cylindrical structures: (**A**) elastic modulus; (**B**) yield strength; (**C**) plateau stress.

**Figure 12 materials-16-03196-f012:**
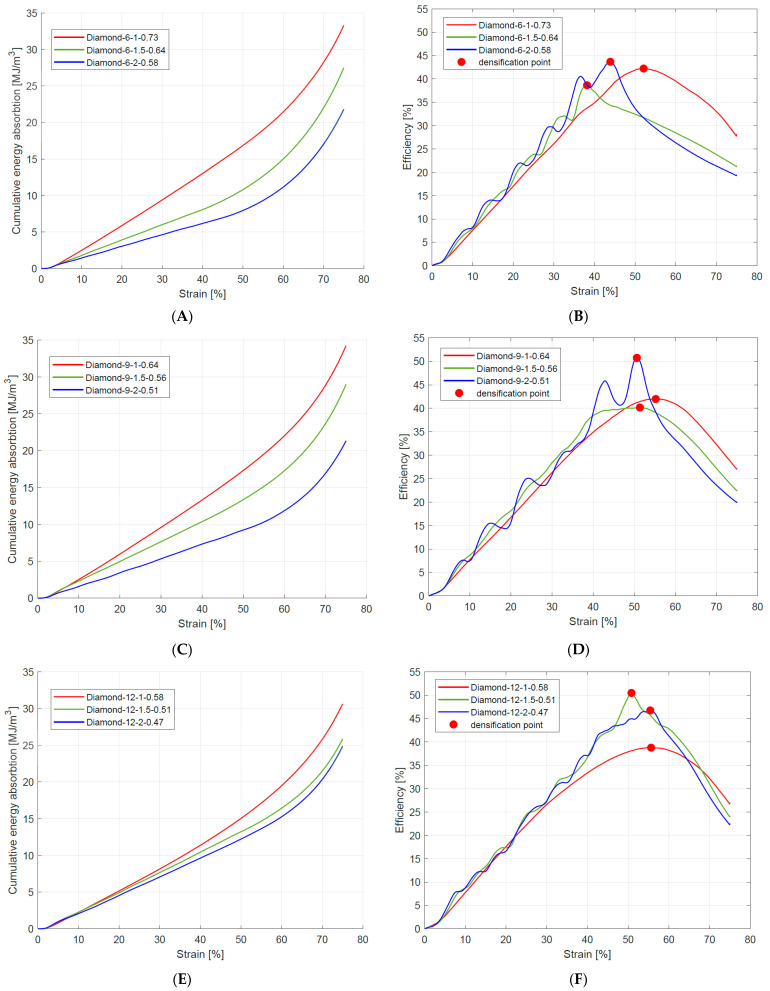
Energy absorption (**A**,**C**,**E**) and energy absorption efficiency (**B**,**D**,**F**) curve of tested Diamond TPMS cylindrical structures.

**Figure 13 materials-16-03196-f013:**
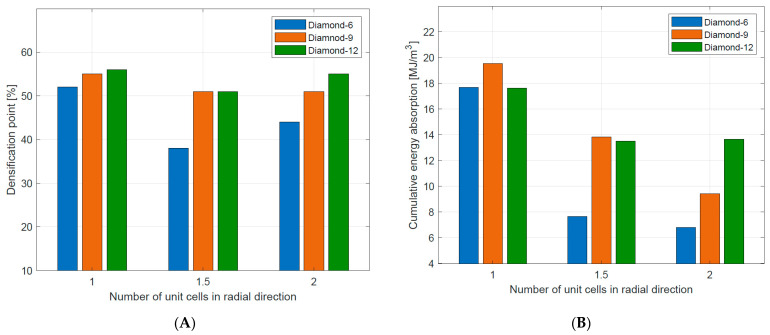
Densification point (**A**) and the energy absorption up to the densification point (**B**) of tested Diamond TPMS cylindrical structures.

**Figure 14 materials-16-03196-f014:**
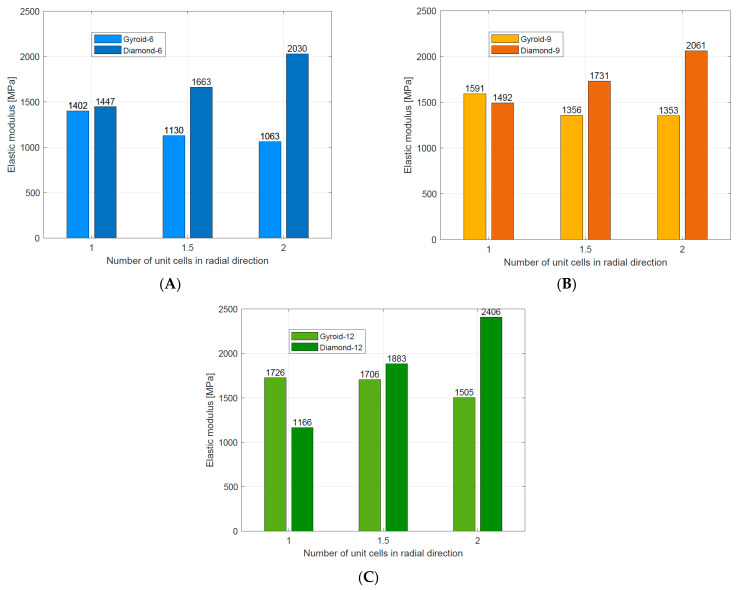
Comparison of elastic modulus for Gyroid [49] and Diamond TPMS cylindrical structures with (**A**) n_circum_= 6; (**B**) n_circum_= 9; (**C**) n_circum_= 12.

**Figure 15 materials-16-03196-f015:**
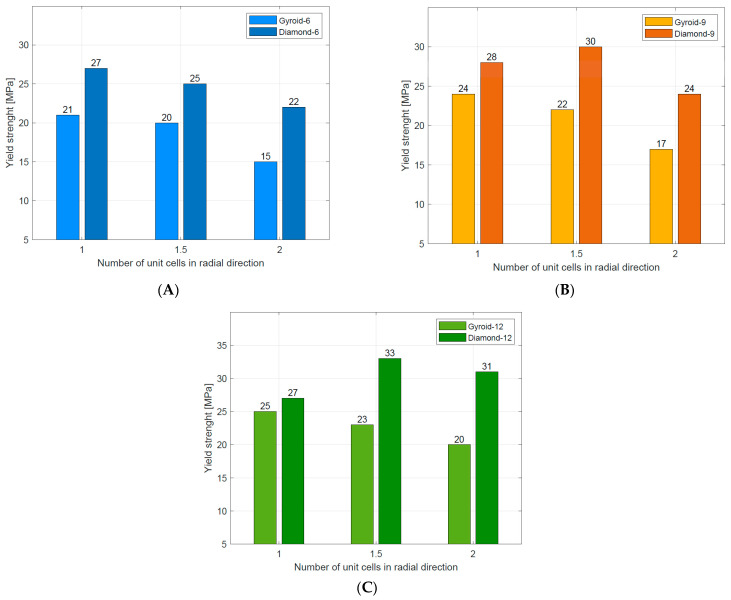
Comparison of yield strength for Gyroid [49] and Diamond TPMS cylindrical structures with (**A**) n_circum_ = 6; (**B**) n_circum_ = 9; (**C**) n_circum_ = 12.

**Figure 16 materials-16-03196-f016:**
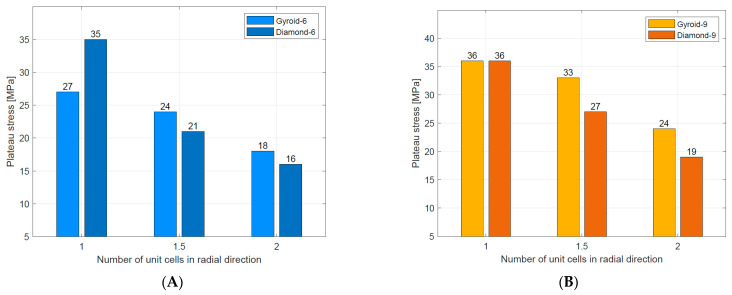
Comparison of plateau stress for Gyroid [49] and Diamond TPMS cylindrical structures with (**A**) n_circum_ = 6; (**B**) n_circum_ = 9; (**C**) n_circum_ = 12.

**Figure 17 materials-16-03196-f017:**
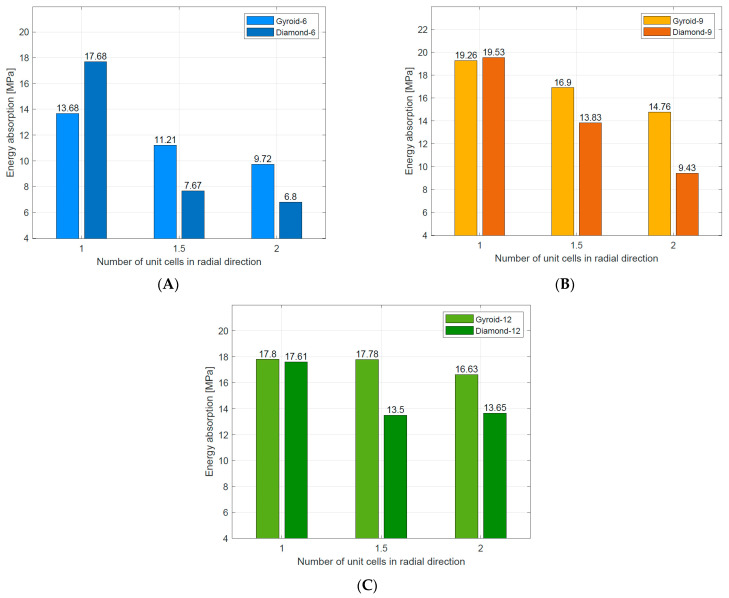
Comparison of energy absorption for Gyroid [49] and Diamond TPMS cylindrical structures with (**A**) n_circum_ = 6; (**B**) n_circum_ = 9; (**C**) n_circum_ = 12.

**Table 2 materials-16-03196-t002:** Parameters of the SLM manufacturing process [52].

Laser Power	Laser Speed	Layer Thickness	Printing Environment
123 W	1000 mm/s	25 µm	Argon

**Table 3 materials-16-03196-t003:** Parameters of sample structures tested.

Symbol	n_circum_	n_radial_	n_axial_	T (mm)	m_r_ (g)	Δm (%)	m_r_avrage_ (g)
Diamond_6_1_0.73					16.42	−0.9	16.43
	6	1	3	0.73	16.45	−0.7	
					16.42	−0.9	
Diamond_6_1.5_0.64					16.46	−0.6	16.47
	6	1.5	3	0.64	16.45	−0.7	
					16.50	−0.4	
Diamond_6_2_0.58					16.47	−0.6	16.46
	6	2	3	0.58	16.45	−0.7	
					16.46	−0.6	
Diamond_9_1_0.64					16.46	−0.6	16.46
	9	1	3	0.64	16.44	−0.8	
					16.47	−0.6	
Diamond_9_1.5_0.56					16.45	−0.7	16.45
	9	1.5	3	0.56	16.45	−0.7	
					16.44	−0.8	
Diamond_9_2_0.51					16.50	−0.4	16.46
	9	2	3	0.51	16.44	−0.8	
					16.44	−0.8	
Diamond_12_1_0.58					16.45	−0.7	16.47
	12	1	3	0.58	16.45	−0.7	
					16.50	−0.4	
Diamond_12_1.5_0.51					16.50	−0.4	16.45
	12	1.5	3	0.51	16.41	−0.9	
					16.43	−0.8	
Diamond_12_2_0.47					16.47	−0.6	16.46
	12	2	3	0.47	16.47	−0.6	
					16.44	−0.8	

**Table 4 materials-16-03196-t004:** Compressive mechanical properties of tested Diamond TPMS cylindrical structures (mean and standard deviation).

Symbol	Elastic Modulus (MPa)	Yield Strength (MPa)	Plateau Stress (MPa)
Diamond_6_1_0.73	1446 ± 162	27 ± 0.33	35 ± 0.30
Diamond_6_1.5_0.64	1663 ± 21	25 ± 0.69	21 ± 0.33
Diamond_6_2_0.58	2030 ± 109	22 ± 0.37	16 ± 0.92
Diamond_9_1_0.64	1493 ± 279	28 ± 0.32	36 ± 0.22
Diamond_9_1.5_0.56	1731 ± 7	30 ± 0.87	27 ± 0.79
Diamond_9_2_0.51	2061 ± 40	24 ± 1.31	19 ± 0.45
Diamond_12_1_0.58	1166 ± 26	27 ± 0.66	30 ± 0.26
Diamond_12_1.5_0.51	1883 ± 142	33 ± 0.87	27 ± 3.68
Diamond_12_2_0.47	2406 ± 180	31 ± 1.49	25 ± 3.93

**Table 5 materials-16-03196-t005:** Basic energy absorption parameters of tested Diamond TPMS cylindrical structures.

Symbol	Densification Point ε_D_ (%)	Energy Absorption W(ε_D_) (MJ/m^3^)
Diamond_6_1_0.73	52	17.68
Diamond_6_1.5_0.64	38	7.67
Diamond_6_2_0.58	44	6.80
Diamond_9_1_0.64	55	19.53
Diamond_9_1.5_0.56	51	13.83
Diamond_9_2_0.51	51	9.43
Diamond_12_1_0.58	56	17.61
Diamond_12_1.5_0.51	51	13.50
Diamond_12_2_0.47	55	13.65

**Table 7 materials-16-03196-t007:** Comparison of maximum energy absorption for Gyroid and Diamond TPMS cylindrical structures.

n_circum_	n_radial_	Maximum Energy Absorption η(ε_D_) (%)		
			Gyroid	Diamond
6	1		45	42
	1.5		38	39
	2		38	44
9	1		45	42
	1.5		40	40
	2		36	51
12	1		42	39
	1.5		38	50
	2		36	47

## Data Availability

Not applicable.

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
