# Peer review of "Mechanical Properties and Energy Absorption Abilities of Diamond TPMS Cylindrical Structures Fabricated by Selective Laser Melting with 316L Stainless Steel"

_materials, 2023, doi:10.3390/ma16083196_

Round 1

Reviewer 1 Report

1. There are differences between stretching-dominated and bending-dominated stress-strain curves. Whether there is a clear critical point between nradial = 1 to 1.5 ?

2. In Figure 8 and 9, the compression mode is overall uniform for nradial = 1, whereas for nradial = 1.5 the compression mode is in a layer-wise manner. What is the intrinsic relationship between the two fracture modes and stress-strain curves? What is the intrinsic relationship between nradial values and fracture patterns? Please explain the reason from the perspective of stress distribution or a more microscopic view.

3. The mechanics of Diamond structures have been studied extensively. Please state the innovation of this article. https://doi.org/10.1016/j.addma.2022.102961. This article may help to improve the quality of the article.

Author Response

Dear Reviewer,

Thank you very much for your comments and suggestion. We have attached a file containing answers to your questions.

Best regards, 

Authors

Reviewer 2 Report

This manuscript focused on the triply periodic minimal surfaces (TPMS) structure with unique energy absorption. The corresponding deformation mechanisms were discussed as well. This study presents the promising potential of mechanical for versatile industrial and civil protection applications. But there are a series of following aspects which need to be further addressed.

1. The interesting mechanical properties of TPMS structure are a consequence of their deliberate structuring, and not only the bulk structures. The authors discuss several parameters of the structures, and it would be better to provide the magnification of each structure to show the inner part of the Diamond TPMS cylindrical ones in Figure 4. 

2. For the quasi-static compression test, it would be better to transform the 2 mm/min to the strain rate based on the dimensions of the specimen. 

3. What is the key parameter in the structure that determines the deformation behavior in TPMS? To this end, the sub-unit structure of the representative triply periodic minimal surfaces should be given to show the deformation traces in the macro-scale structures. For example, in Figure 8 and Figure 9, only the deformation behavior in the macroscale were shown, the deformation in the inner part should also be given. The authors can find the approach mentioned in this reference Yang, Ting, et al. "High strength and damage-tolerance in echinoderm stereom as a natural bicontinuous ceramic cellular solid." Nature Communications 13.1 (2022): 6083.

4. In the Figure 5 to Figure 7, the peak stress appears at the strain around 3%~5% in the stress-strain curves. So, what is the deformation behavior after peak stress between 0% to 20% strain level? Therefore in Fig. 9, the deformation mode between 0% to 20% strain level should be provided, as well. 

5. Why the red curves in Fig.5 and Fig.6 are smoother than the other two ? The author can explain it in more detail. In addition, there is mistake in Line 221 Bast on this.   

6. The homogeneous structure was studied, and it would be interesting to prepare the gradient microstructures, and investigate the post deformation behavior in the TPMS structure. 

Author Response

Dear Reviewer,

Thanh you very much for your comments and suggestions. We have attached a file containing answers to your questions.

Best regards, 

Authors

Reviewer 3 Report

In the review of the manuscript entitled Mechanical Properties and Energy Absorption Abilities of Diamond TPMS Cylindrical Structures Fabricated by Selective Laser Melting with 316L Stainless Steel. The authors have provided a good description and the methodology is also fine. I would like to see this article publish but after some questions as follow;

1.     What is the significance of using 316L stainless steel in the fabrication of diamond TPMS cylindrical structures?

2.     How were the diamond TPMS cylindrical structures fabricated using selective laser melting? Can you explain the process in detail?

3.     What is the effect of different process parameters such as laser power, scan speed, and hatch spacing on the mechanical properties of the structures?

4.     How were the mechanical properties of the structures characterized? What techniques were used for testing and analysis?

5.     Can you explain the results of the energy absorption tests performed on the structures? How do they compare to other materials commonly used for energy absorption applications?

6.     What are the potential applications of diamond TPMS cylindrical structures in industries such as aerospace and automotive?

7.     What are some of the limitations of the current study and what are some future research directions that can build upon this work?

Author Response

Dear Reviewer,

Thank you very much for your comments and suggestions. We have attached a file containing answers to your questions.

Best regards,

Authors
